# RIEMANNIAN MULTICLASS LOGISTICS REGRESSION FOR SPD NEURAL NETWORKS

## ABSTRACT

Deep neural networks for learning Symmetric Positive Definite (SPD) matrices are gaining increasing attention in machine learning. Despite the significant progress, most existing SPD networks use traditional Euclidean classifiers on approximated spaces rather than intrinsic classifiers that accurately capture the geometry of SPD manifolds. Inspired by the success of Hyperbolic Neural Networks (HNNs), we propose Riemannian multiclass logistics regression (RMLR) for the classification layers in SPD networks. We focus on the metrics pulled back from the Euclidean space, such as Log-Euclidean Metric (LEM) and Log-Cholesky Metric (LCM), and introduce a unified framework for building Riemannian classifiers under these metrics. We first generalize the existing LEM and LCM by the concept of deformation and then design the specific SPD classifiers. Our framework encompasses the most popular LogEig classifier in existing SPD networks as a special case. The effectiveness of our method is demonstrated in three applications: radar recognition, human action recognition, and electroencephalography (EEG) classification.

## 1 INTRODUCTION

Symmetric Positive Definite (SPD) matrices are commonly encountered in a diverse range of scientific fields, such as medical imaging (Chakraborty et al., 2018; 2020), signal processing (Arnaudon et al., 2013; Hua et al., 2017; Brooks et al., 2019b;a), elasticity (Moakher, 2006; Guilleminot & Soize, 2012), question answering (López et al., 2021; Nguyen, 2022a), and computer vision (Huang & Van Gool, 2017; Harandi et al., 2018; Zhen et al., 2019; Chakraborty, 2020; Zhang et al., 2020; Chakraborty, 2020; Song et al., 2021; Nguyen, 2021; 2022b; Song et al., 2022b). Despite their ubiquitous presence, traditional learning algorithms are ineffective in handling the non-Euclidean geometry of SPD matrices. To address this limitation, several Riemannian metrics have been proposed, including Affine-Invariant Metric (AIM) (Pennec et al., 2006), Log-Euclidean Metric (LEM) (Arsigny et al., 2005), and Log-Cholesky Metric (LCM) (Lin, 2019). With these Riemannian metrics, various machine learning techniques can be generalized into SPD manifolds.

Inspired by the great success of deep learning (Hochreiter & Schmidhuber, 1997; Krizhevsky et al., 2012; He et al., 2016), several deep networks have been developed on SPD manifolds. Despite their promising performance, many approaches still rely on Euclidean spaces for classification, such as tangent spaces (Huang & Van Gool, 2017; Brooks et al., 2019a; Nguyen, 2021; Wang et al., 2021; Nguyen, 2022a;b; Kobler et al., 2022; Wang et al., 2022; Chen et al., 2023b), ambient Euclidean spaces (Wang et al., 2020; Song et al., 2021; 2022a), and coordinate systems (Chakraborty et al., 2018). However, these strategies distort the intrinsic geometry of the SPD manifold, undermining the effectiveness of SPD neural networks. Recently, motivated by HNNs (Ganea et al., 2018), Nguyen & Yang (2023) developed three kinds of SPD Multiclass Logistics Regression (MLR) based on the gyro-structures induced by LEM, LCM and AIM. However, the proposed SPD MLRs in Nguyen & Yang (2023) rely on the gyro-structures, limiting their generality. Chakraborty et al. (2020) also introduced an invariant layer for manifold-valued data that mimics the invariant FC layer in CNNs. However, it is designed for gridded manifold-valued data, which is not the primary data type encountered in many other SPD networks. Following the convention of most SPD networks, we only focus on non-gridded cases.

*In fact, SPD MLR can be directly derived under LEM and LCM without the assistance of gyro structures.* More generally, LEM and LCM belong to Pullback Euclidean Metrics (PEMs), which are metrics pulled back from the Euclidean space. In this paper, we focus on PEMs and propose a unified framework for building SPD Multiclass Logistics Regression (SPD MLR) under PEMs.

On the empirical side, we first generalize the existing LEM and LCM into parameterized metric families through the concept of deformation. We then showcase our SPD MLRs under these generalized metrics. Besides, our framework encompasses the gyro SPD MLRs induced by LEM and LCM in Nguyen & Yang (2023). More importantly, our framework also provides an intrinsic explanation for the commonly used LogEig classifier on SPD manifolds, consisting of successive matrix logarithm, FC, and softmax layers. Finally, extensive experiments demonstrate that our proposed Riemannian classifiers exhibit consistent performance gains across widely used SPD benchmarks. The **contributions** of our work are summarized as follows:

(a) We introduce a general framework for building SPD MLRs under PEMs and design specific SPD MLRs under two parameterized metric families.

(b) Our framework offers an intrinsic explanation of the most popular LogEig classifier.

(c) Extensive experiments on widely used SPD learning benchmarks demonstrate the superiority of our proposed classifiers over the previous baselines.

**Main theoretical results:** In Defs. 3.1 and 3.2, we introduce the definitions of the SPD hyperplane and SPD MLR, respectively. The core idea lies in the computation of marginal distance to the hyperplane defined in Eq. (14). As Lem. 3.3 demonstrates, this problem admits a closed-form solution under any PEM. Consequently, we establish the uniform expression of SPD MLR under any PEM in Thm. 3.6. Sec. 4.1 details the extension of existing LEM and LCM by the concept of deformation. Prop. 4.1 shows that the deformed LEM and LCM all belong to PEMs, while their deformation utility is discussed in Prop. 4.2. Expressions for SPD MLRs under the deformed LEM and LCM are presented in Cor. 4.3. Finally, our framework also offers an intrinsic explanation for the widely used LogEig classifier in Prop. 5.1. Due to page limits, all the proofs are placed in App. D.

## 2 GEOMETRY OF SPD MANIFOLDS

In this section, we briefly review some basic concepts in differential geometry and SPD manifolds. For in-depth discussions, please refer to Do Carmo & Flaherty Francis (1992); Tu (2011).

We first recap the concept of the pullback metric, which is ubiquitous in differential manifolds.

**Definition 2.1** (Pullback Metrics). Suppose $\mathcal{M}, \mathcal{N}$ are smooth manifolds, $g$ is a Riemannian metric on $\mathcal{N}$, and $f : \mathcal{M} \to \mathcal{N}$ is a diffeomorphism. Then $f$ can induce a Riemannian metric on $\mathcal{M}$ defined as

$$(f^*g)_p(V_1, V_2) = g_{f(p)}(f_{*,p}(V_1), f_{*,p}(V_2)), \tag{1}$$

where $p \in \mathcal{M}$, $f_{*,p}(\cdot)$ is the differential map of $f$ at $p$, and $V_i \in T_p\mathcal{M}$. We call $f^*g$ as the pullback metric by $f$ from $\mathcal{N}$.

Now, we introduce some necessary preliminaries about SPD manifolds. The set of SPD matrices, denoted as $\mathcal{S}_{++}^n$, forms a smooth manifold known as the SPD manifold (Arsigny et al., 2005). Several successful Riemannian metrics have been established on SPD manifolds, such as LEM (Arsigny et al., 2005), AIM (Pennec et al., 2006) and LCM (Lin, 2019). Recently, Thanwerdas & Pennec (2023) generalized LEM and AIM into two-parameter families of metrics, namely $(\alpha, \beta)$-AIM and $(\alpha, \beta)$-LEM by the O(n)-invariant inner product on the Euclidean space $\mathcal{S}^n$ of symmetric matrices:

$$\langle V, W \rangle^{(\alpha, \beta)} = \alpha \langle V, W \rangle + \beta \operatorname{tr}(V) \operatorname{tr}(W), \text{ with } (\alpha, \beta) \in \mathbf{ST}, \tag{2}$$

where $\mathbf{ST} = \{(\alpha, \beta) \in \mathbb{R}^2 \mid \min(\alpha, \alpha + n\beta) > 0\}$, and $V, W \in \mathcal{S}^n$.

In this study, we focus on $(\alpha, \beta)$-LEM and LCM. We first make some notations and then summarize all the necessary Riemannian operators in Tab. 1. Given SPD matrices $P, Q \in \mathcal{S}_{++}^n$ along with tangent vectors $V, W \in T_P\mathcal{S}_{++}^n$, we introduce the following notations. Specifically, the Riemannian metric at $P$ is represented as $g_P(\cdot, \cdot)$, while $\operatorname{Log}_P(\cdot)$ denotes the Riemannian logarithm at $P$. $\Gamma_{P \to Q}$ signifies the parallel transport along the geodesic connecting $P$ and $Q$. The matrix exponential and logarithmic functions are denoted as $\operatorname{mexp}(\cdot)$ and $\operatorname{mlog}(\cdot)$, respectively. In addition, $\operatorname{Chol}(\cdot)$ denotes the Cholesky decomposition, with $L = \operatorname{Chol} P$ and $K = \operatorname{Chol} Q$ representing the Cholesky factors of $P$ and $Q$. The differentials of $\operatorname{mlog}$ and $\operatorname{Chol}^{-1}$ at $P$ and $L$ are respectively denoted as $\operatorname{mlog}_{*,P}$ and $(\operatorname{Chol})_{*,L}^{-1}$. $\lfloor \cdot \rfloor$ refers to the strictly lower part of a square matrix, and $\operatorname{Dlog}(L)$ denotes a diagonal matrix comprised of the logarithm of the diagonal elements of $L$.

Table 1: Riemannian operators of $(\alpha, \beta)$-LEM and LCM on SPD manifolds.

| Name | $g_P(V, W)$ | $\mathrm{Log}_P Q$ | $\Gamma_{P \to Q}(V)$ |
|---|---|---|---|
| $(\alpha, \beta)$-LEM | $\langle \mathrm{mlog}_{*,P}(V), \mathrm{mlog}_{*,P}(W) \rangle^{(\alpha, \beta)}$ | $(\mathrm{mlog}_{*,P})^{-1}[\mathrm{mlog}(Q) - \mathrm{mlog}(P)]$ | $(\mathrm{mlog}_{*,Q})^{-1} \circ \mathrm{mlog}_{*,P}(V)$ |
| LCM | $\sum_{i>j} \tilde{V}_{ij}\tilde{W}_{ij} + \sum_{j=1}^n \tilde{V}_{jj}\tilde{W}_{jj}L_{jj}^{-2}$ | $(\mathrm{Chol}^{-1})_{*,L}\left[\lfloor K \rfloor - \lfloor L \rfloor + \mathbb{D}(L)\,\mathrm{Dlog}(\mathbb{D}(L)^{-1}\mathbb{D}(K))\right]$ | $(\mathrm{Chol}^{-1})_{*,K}\left[\lfloor \tilde{V} \rfloor + \mathbb{D}(K)\mathbb{D}(L)^{-1}\mathbb{D}(\tilde{V})\right]$ |

Following the terminology in Chen et al. (2023a), we define the pullback metrics from Euclidean spaces by diffeomorphisms as the Pullback Euclidean Metrics (PEMs). Chen et al. (2023a) demonstrated that both LEM and LCM are PEMs. We recall an excerpt from Theorem 4.2 of Chen et al. (2023a), covering the properties of PEMs on SPD manifolds.

**Theorem 2.2** (Pullback Euclidean Metrics (PEMs)). *Let* $S_1, S_2 \in \mathcal{S}_{++}^n$, $\phi : \mathcal{S}_{++}^n \to \mathcal{S}^n$ *is a diffeomorphism. We define the following operations,*

$$S_1 \odot_\phi S_2 = \phi^{-1}(\phi(S_1) + \phi(S_2)), \tag{3}$$

$$g_S^\phi(V_1, V_2) = \langle \phi_{*,S}(V_1), \phi_{*,S}(V_2) \rangle, \forall S \in \mathcal{S}_{++}^n, \forall V_i \in T_S \mathcal{S}_{++}^n, \tag{4}$$

*where* $\phi_{*,S} : T_S \mathcal{S}_{++}^n \to T_{\phi(S)} \mathcal{S}^n$ *is the differential map of* $\phi$ *at* $S$, *and* $\langle \cdot, \cdot \rangle$ *is the standard Frobenius inner product. Then, we have the following conclusions:* $\{\mathcal{S}_{++}^n, \odot_\phi\}$ *is an Abelian Lie group,* $\{\mathcal{S}_{++}^n, g^\phi\}$ *is a Riemannian manifold, and* $g^\phi$ *is a bi-invariant metric, called Pullback Euclidean Metric (PEM). The associated geodesic distance is*

$$d^\phi(S_1, S_2) = \|\phi(S_1) - \phi(S_2)\|_{\mathrm{F}}, \tag{5}$$

*where* $\| \cdot \|_{\mathrm{F}}$ *is the norm induced by* $\langle \cdot, \cdot \rangle$. *The Riemannian operators are as follows*

$$\mathrm{Exp}_{S_1} V = \phi^{-1}(\phi(S_1) + \phi_{*,S_1} V), \tag{6}$$

$$\mathrm{Log}_{S_1} S_2 = \phi_{*,\phi(S_1)}^{-1}(\phi(S_2) - \phi(S_1)), \tag{7}$$

$$\Gamma_{S_1 \to S_2}(V) = \phi_{*,\phi(S_2)}^{-1} \circ \phi_{*,S_1}(V), \tag{8}$$

*where* $V \in T_{S_1} \mathcal{S}_{++}^n$ *is a tangent vector,* $\mathrm{Exp}_{S_1}$ *is the Riemannian exponential at* $S_1$, *and* $\phi_*^{-1}$ *are the differential maps* $\phi^{-1}$.

## 3 SPD MLR ON SPD MANIFOLDS

This section first reformulates the Euclidean MLR. Then, we deal with SPD MLR under arbitrary PEM on SPD manifolds.

### 3.1 REFORMULATION OF EUCLIDEAN MLR

Lebanon & Lafferty (2004) first reformulated the Euclidean MLR from the perspective of distances to margin hyperplanes. Hyperbolic MLR was designed based on this reformulation (Ganea et al., 2018). Nguyen & Yang (2023) further proposed three gyro SPD MLRs based on the gyro-structures induced by AIM, LEM, and LCM. We now briefly review the reformulation of Euclidean MLR.

Given $C$ classes, MLR in $\mathbb{R}^n$ computes the follwoing softmax probabilities:

$$\forall k \in \{1, \ldots, C\}, \quad p(y = k \mid x) \propto \exp\left((\langle a_k, x \rangle - b_k)\right), \quad \text{where } b_k \in \mathbb{R}, x, a_k \in \mathbb{R}^n. \tag{9}$$

As shown in (Lebanon & Lafferty, 2004, Sec. 5) and (Ganea et al., 2018, Sec. 3.1), Eq. (9) can be reformulated as

$$p(y = k \mid x) \propto \exp(\mathrm{sign}(\langle a_k, x - p_k \rangle)\|a_k\|d(x, H_{a_k, p_k})), p_k, x \in \mathbb{R}^n, \text{ and } a_k \in \mathbb{R}^n \backslash \{\mathbf{0}\}, \tag{10}$$

$$H_{a,p} = \{x \in \mathbb{R}^n : \langle a, x - p \rangle = 0\}, \quad \text{where } a \in \mathbb{R}^n \backslash \{\mathbf{0}\}, \text{ and } p \in \mathbb{R}^n, \tag{11}$$

where $H_{a,p}$ is referred to a hyperplane.

In geometry, $\mathrm{Log}_p x$ is the natural generalization of the directional vector $\vec{px} = x - p$ starting at $p$ and ending at $x$. The Riemannian metric at $p$ can also replace the inner product. More detail can be found in Pennec et al. (2006, Table 1). Therefore, the MLR in Eq. (10) and hyperplane in Eq. (11) can be readily generalized into SPD manifolds $\{\mathcal{S}_{++}^n, g\}$.

**Definition 3.1** (SPD hyperplanes)**.** Given $P \in \mathcal{S}_{++}^n, A \in T_P\mathcal{S}_{++}^n\backslash\{\mathbf{0}\}$, we define the SPD hyperplane as

$$\tilde{H}_{A,P} = \{S \in \mathcal{S}_{++}^n : g_P(\mathrm{Log}_P S, A) = \langle \mathrm{Log}_P S, A \rangle_P = 0\}, \tag{12}$$

where $P$ and $A$ are referred to as shift and normal matrices, respectively.

It can be proven that on geodesic complete manifolds, the SPD hyperplanes are submanifolds (App. C). Nevertheless, we still follow the nomenclature of Ganea et al. (2018); Lebanon & Lafferty (2004) and call $\tilde{H}_{A,P}$ SPD hyperplane.

**Definition 3.2** (SPD MLR)**.** SPD MLR is defined as

$$p(y = k \mid S) \propto \exp(\mathrm{sign}(\langle A_k, \mathrm{Log}_{P_k}(S)\rangle_{P_k})\|A_k\|_{P_k} d(S, \tilde{H}_{A_k,P_k})), \tag{13}$$

where $P_k \in \mathcal{S}_{++}^n$, $A_k \in T_{P_k}\mathcal{S}_{++}^n\backslash\{\mathbf{0}\}$, $\langle \cdot, \cdot \rangle_{P_k} = g_{P_k}$, and $\| \cdot \|_{P_k}$ is the norm on $T_{P_k}\mathcal{S}_{++}^n$ induced by $g$ at $P_k$, and $\tilde{H}_{A_k,P_k}$ is a margin hyperplane in $\mathcal{S}_{++}^n$ as defined in Eq. (12). $d(S, \tilde{H}_{A_k,P_k})$ denotes the margin distance between $S$ and SPD hyperplane $\tilde{H}_{A_k,P_k}$, which is formulated as:

$$d(S, \tilde{H}_{A_k,P_k})) = \inf_{Q \in \tilde{H}_{A_k,P_k}} d(S, Q), \tag{14}$$

where $d(S, Q)$ is the geodesic distance induced by $g$.

**Difference with the gyro SPD MLR:** Although the gyro SPD MLR introduced in Nguyen & Yang (2023) and our method both extend the Euclidean MLR into SPD manifolds, there exist two main differences:

1. **The mathematical techniques employed are different.** Nguyen & Yang (2023) adopted gyro structures to reformulate Eqs. (10) and (11). However, their gyro structures are induced by the Riemannian metrics. Also, the gyro inner product and gyro norm (Nguyen & Yang, 2023, Def. 2.15) are defined by the inner product and norm in the tangent space at the identity matrix, *i.e.,* $T_I\mathcal{S}_{++}^n$. In contrast, our approach directly applies Riemannian geometry to reformulate Euclidean MLR.

2. **The margin distance in Eq. (14) are calculated differently.** The margin distance in gyro SPD MLR shares the same expression as our Eq. (14), except that the distance in the right-hand side is gyro distance, which is defined by the distance on $T_I\mathcal{S}_{++}^n$. To bypass the optimization problem in Eq. (14), Nguyen & Yang (2023) introduced the *pseudo-gyrodistance* and pointed out that under the specific LEM and LCM, the pseudo-gyrodistance is equal to the margin distance. In contrast, we directly use the geodesic distance, which is the most natural descriptor for characterizing the distance on manifolds.

### 3.2 SPD MLRs under PEMs

Recalling that for our SPD MLR in Def. 3.2, under most Riemannian metrics on SPD manifolds, all the involved operators in Eq. (13) have close form expressions, except the margin distance in Eq. (14). Therefore, the only difficulty lies in the calculation of the margin distance. This subsection follows the notations in Thm. 2.2 and proposes a general expression for SPD MLRs under PEMs.

We chose PEMs as our starting metrics mainly because of its extensive inclusion and easy computation. Several Riemannian metrics, including LEM, LCM, and their variants Thanwerdas & Pennec (2023; 2022), all end up as PEMs. Besides, due to the fast and simple calculation of PEMs, the margin distance under the PEM has a closed-form expression, while other metrics like AIM would be complicated to obtain the distances to hyperplanes.

We start by calculating the margin distance in Eq. (14) under a given PEM.

**Lemma 3.3.** *Under a Riemannian metric $g$ which belongs to PEMs, the margin distance defined in Eq.* (14) *has a closed-form solution:*

$$d(S, \tilde{H}_{A_k,P_k})) = d(\phi(S), H_{\phi_{*,P_k}(A_k),\phi(P_k)}) = \frac{|\langle \phi(S) - \phi(P_k), \phi_{*,P_k}(A_k)\rangle|}{\|A_k\|_{P_k}}, \tag{15}$$

*where $|\cdot|$ is the absolute value.*

Putting Eq. (15) into Eq. (13), we obtain our SPD MLR under a given PEM:

$$p(y = k \mid S) \propto \exp(\langle A_k, \mathrm{Log}_{P_k}(S)\rangle_{P_k}) = \exp(\langle \phi(S) - \phi(P_k), \phi_{*,P_k}(A_k)\rangle), \qquad (16)$$

where $S, P_k \in \mathcal{S}_{++}^n$ and $A_k \in T_{P_k}\mathcal{S}_{++}^n \backslash \{\mathbf{0}\}$. When $P_k$ is fixed, $A_k \in T_{P_k}\mathcal{S}_{++}^n$ indeed lies in a Euclidean space. However, $P_k$ would vary during training, making $A_k$ non-Euclidean. To remedy this issue, we propose two solutions. The first one is the parallel transportation from a fixed tangent space, writing $A_k = \Gamma_{Q \to P_k}(\tilde{A}_k)$ with $\tilde{A}_k \in T_Q\mathcal{S}_{++}^n$ as a Euclidean parameter. This is the solution also adopted by HNNs (Ganea et al., 2018), where the tangent point is the zero vector. Alternatively, one can also rely on the differential of a Lie group translation, which is widely used in differential manifolds (Tu, 2011, § 20). Since the Lie groups associated with PEMs are Abelian, we only consider the left translation in the paper. We have the following two lemmas to show the relation between the parallel transport and the differential of left translation.

**Lemma 3.4.** *Under a given PEM, any parallel transportation is equivalent to the differential map of a left translation and vice versa.*

**Lemma 3.5.** *Given two fixed SPD matrices $Q_1, Q_2 \in \mathcal{S}_{++}^n$, we have the following equivalence for parallel transportations under a PEM,*

$$\forall \tilde{A}_{1,k} \in T_{Q_1}\mathcal{S}_{++}^n, \exists! \tilde{A}_{2,k} \in T_{Q_2}\mathcal{S}_{++}^n, s.t. \Gamma_{Q_1 \to P_k}(\tilde{A}_{1,k}) = \Gamma_{Q_2 \to P_k}(\tilde{A}_{2,k}). \qquad (17)$$

Lem. 3.4 indicates that under PEMs, the above two solutions are equivalent, while Lem. 3.5 implies that anchor points can be arbitrarily chosen. Therefore, without loss of generality, we generate $A_k$ from the tangent space at the identity matrix $I$ by parallel transportation, *i.e.*, $A_k = \Gamma_{I \to P_k}(\tilde{A}_k)$ with $\tilde{A}_k \in T_I\mathcal{S}_{++}^n \cong \mathcal{S}^n$. Together with Eq. (8), Eq. (16) can be further simplified.

**Theorem 3.6** (SPD MLR under a PEM). *Under any PEM, SPD MLR and SPD hyperplane is*

$$p(y = k \mid S) \propto \exp(\langle \phi(S) - \phi(P_k), \phi_{*,I}(\tilde{A}_k)\rangle), \qquad (18)$$

$$\tilde{H}_{\tilde{A}_k, P_k} = \{S \in \mathcal{S}_{++}^n : \langle \phi(S) - \phi(P_k), \phi_{*,I}(\tilde{A}_k)\rangle = 0\}, \qquad (19)$$

*where $\tilde{A}_k \in T_I\mathcal{S}_{++}^n/\{0\} \cong \mathcal{S}^n$ is a symmetric matrix, and $P_k \in \mathcal{S}_{++}^n$ is an SPD matrix.*

## 4 SPD MLRs under the deformed LEM and LCM

In this section, we first generalize the existing LEM and LCM by the idea of deformation, and then we showcase our SPD MLR in Thm. 3.6 under these generalized metrics.

### 4.1 Deformed LEM and LCM on SPD manifolds

Inspired by the deforming utility of the matrix power function (Thanwerdas & Pennec, 2019; 2022), this subsection introduces the deformed $(\alpha, \beta)$-LEM and LCM. We define $(\theta, \alpha, \beta)$-LEM as the pullback metric of $(\alpha, \beta)$-LEM by matrix power function $(\cdot)^\theta$ and scaled by $\frac{1}{\theta^2}$. Similarly, $(\theta)$-LCM is the pullback metric of LEM by matrix power function $(\cdot)^\theta$ and scaled by $\frac{1}{\theta^2}$. As both the standard LEM and LCM are PEMs, it can be expected that $(\theta, \alpha, \beta)$-LEM and $(\theta)$-LCM are also PEMs.

**Proposition 4.1.** *For all $\theta \neq 0$ and $(\alpha, \beta) \in \mathbf{ST}$, $(\theta, \alpha, \beta)$-LEM and $(\theta)$-LCM belongs to PEMs.*

The parameter $\theta$ in $(\theta, \alpha, \beta)$-LEM and $(\theta)$-LCM also serves as deformation. We have the following proposition for the deforming utility of $\theta$ in these two metrics.

**Proposition 4.2** (Deformation). *$(\theta, \alpha, \beta)$-LEM interpolates between $(\alpha, \beta)$-LEM ($\theta \to 0$) and $(\alpha, \beta)$-LEM ($\theta = 1$). $\theta$-LCM interpolates between $\tilde{g}$-LEM ($\theta = 0$) and LCM ($\theta = 1$), with $\tilde{g}$-LEM defined as*

$$\langle V, W \rangle_P = \tilde{g}(\mathrm{mlog}_{*,P}(V), \mathrm{mlog}_{*,P}(W)), \forall P \in \mathcal{S}_{++}^n, \forall V, W \in T_P\mathcal{S}_{++}^n, \qquad (20)$$

*where $\tilde{g}(V_1, V_2) = \frac{1}{2}\langle V_1, V_2 \rangle - \frac{1}{4}\langle \mathbb{D}(V_1), \mathbb{D}(V_2) \rangle$, $\mathbb{D}(V_i)$ is a diagonal matrix consisting of the diagonal elements of $V_i$, and $\mathrm{mlog}_{*,P}$ is the differential map at $P$.*

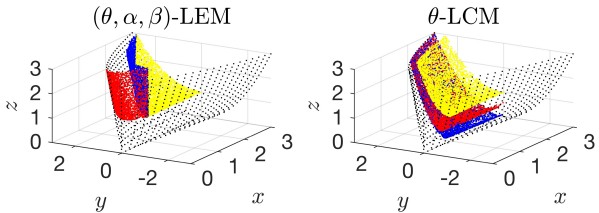

Figure 1: Conceptual illustration of SPD hyperplanes induced by $(\theta, \alpha, \beta)$-LEM and $(\theta)$-LCM. In each subfigure, the black dots are symmetric positive semi-definite (SPSD) matrices, denoting the boundary of $\mathcal{S}_{++}^2$, while the blue, red, and yellow dots denote three SPD hyperplanes.

### 4.2 SPD MLRs under the Deformed LEM and LCM

Since both $(\theta, \alpha, \beta)$-LEM and $(\theta)$-LCM are PEMs, the SPD MLRs under these two families of metrics can be directly obtained by Thm. 3.6.

**Corollary 4.3** (SPD MLRs under the deformed LEM and LCM). *The SPD MLRs under* $(\theta, \alpha, \beta)$-*LEM is*

$$p(y = k \mid S) \propto \exp\left[\theta^2 \langle \mathrm{mlog}(S) - \mathrm{mlog}(P_k), \tilde{A}_k \rangle^{(\alpha,\beta)}\right], \tag{21}$$

*where* $\tilde{A}_k \in T_I \mathcal{S}_{++}^n \cong \mathcal{S}^n$ *and* $P_k \in \mathcal{S}_{++}^n$. *The SPD MLRs under* $(\theta)$-*LCM is*

$$p(y = k \mid S) \propto \exp\left[\mathrm{sgn}(\theta)\langle \lfloor \tilde{K} \rfloor - \lfloor \tilde{L} \rfloor + \sqrt{|\theta|}\left[\mathrm{Dlog}(\tilde{K}) - \mathrm{Dlog}(\tilde{L})\right], \lfloor \tilde{A}_k \rfloor + \frac{\sqrt{|\theta|}}{2}\mathbb{D}(\tilde{A}_k)\rangle\right], \tag{22}$$

*where* $\tilde{K} = \mathrm{Chol}(S^\theta)$, $\tilde{L} = \mathrm{Chol}(P_k^\theta)$, *and* $\mathbb{D}(\tilde{A}_k)$ *denotes a diagonal matrix with diagonal elements of* $\tilde{A}_k$.

$\mathcal{S}_{++}^2$ can be visualized in $\mathbb{R}^3$ by the condition that $\forall P = \begin{pmatrix} x & y \\ y & z \end{pmatrix} \in \mathcal{S}^2$ is positive definite iff $x, z > 0 \land xz > y^2$. Fig. 1 illustrates SPD hyperplanes induced by $(\theta, \alpha, \beta)$-LEM and $(\theta)$-LCM.

*Remark* 4.4. Our SPD MLR incorporates the gyro SPD MLRs induced by LEM and LCM presented in Nguyen & Yang (2023). For $(\theta, \alpha, \beta)$-LEM, when $(\theta, \alpha, \beta) = (1, 1, 0)$, $(\theta, \alpha, \beta)$-LEM becomes the standard LEM. Our SPD MLR in Eq. (21) becomes the gyro SPD MLR induced by LEM (Nguyen & Yang, 2023, Thm. 2.23). For $(\theta)$-LCM, when $\theta = 1$, the $(\theta)$-LCM becomes the standard LCM. Our SPD MLR in Eq. (22) becomes the gyro SPD MLR induced by LCM (Nguyen & Yang, 2023, Thm. 2.24). However, our framework does not require gyro structures and directly builds SPD MLR based on the Riemannian metric.

## 5 Understanding the Existing LogEig Classifier

Many of the existing SPD neural networks (Huang & Van Gool, 2017; Brooks et al., 2019a; Nguyen et al., 2019; Wang et al., 2021; Nguyen, 2021; Wang et al., 2022; Chen et al., 2023b) rely on a Euclidean MLR in the codomain of matrix logarithm, *i.e.,* matrix logarithm followed by an FC layer and a softmax layer. For simplicity, we call this classifier as LogEig MLR. The existing explanation of LogEig MLR is approximating manifolds by tangent space. However, the widely used LogEig MLR can be geometrically explained as a particular case of our approach.

When $(\theta, \alpha, \beta) = (1, 1, 0)$ for $(\theta, \alpha, \beta)$-LEM, the SPD MLR in Eq. (21) is very similar to the LogEig MLR. However, due to the nonlinearity of $\mathrm{mlog}(\cdot)$ and the non-Euclideanness of SPD parameter $P_k$, SPD MLR cannot be hastily viewed as equivalent to LogEig MLR. Nevertheless, under special circumstances, Eq. (21) is equivalent to a LogEig MLR.

**Proposition 5.1.** *Endowing SPD manifolds with the standard LEM, optimizing SPD parameter $P_k$ in (21) by LEM-based RSGD and Euclidean parameter $A_k$ by Euclidean SGD, the LEM-based SPD MLR is equivalent to a LogEig MLR with parameters in FC layer optimized by Euclidean SGD.*

Prop. 5.1 implies that optimized by LEM-based RSGD, the LEM-based SPD MLR is equivalent to the Euclidean MLR in the codomain of matrix logarithm. Nevertheless, a substantial body of prior works underscores the theoretical and empirical superiority of the AIM-based optimization over its LEM-based counterpart (Sra & Hosseini, 2015; Han et al., 2021). Therefore, we also adopt the AIM-based optimizer in this paper to update the involved SPD parameters.

## 6 EXPERIMENTS

In this section, we implement the proposed two families of SPD MLRs to SPD neural networks. Note that our SPD MLRs are architecture-agnostic and can be applied to any existing SPD neural network. This paper focuses on two network architectures, SPDNet (Huang & Van Gool, 2017) and TSMNet+SPDDSMBN (Kobler et al., 2022). SPDNet is the most classic SPD neural network. Following previous works (Huang & Van Gool, 2017; Brooks et al., 2019a), we evaluate our SPD MLRs under this architecture for radar recognition on the Radar dataset (Brooks et al., 2019a) and human action recognition on the HDM05 (Müller et al., 2007). TSMNet+SPDDSMBN is the SOTA Riemannian approach to EEG classification, which is the improved version of SPDNetBN (Brooks et al., 2019a) for transfer learning on EEG tasks. We evaluate our SPD MLRs under this baseline for EEG classification on the Hinss2021 dataset (Hinss et al., 2021).

**Baseline models:** SPDNet (Huang & Van Gool, 2017) mimics the conventional densely connected feedforward network, consisting of three basic building blocks

$$\text{BiMap layer: } S^k = W^{k\top} S^{k-1} W^k, \text{ with } W^k \text{ semi-orthogonal,} \tag{23}$$

$$\text{ReEig layer: } S^k = U^{k-1} \max(\Sigma^{k-1}, \epsilon I_n) U^{k-1\top}, \text{ with } S^{k-1} = U^{k-1}\Sigma^{k-1}U^{k-1\top}, \tag{24}$$

$$\text{LogEig layer: } S^k = \text{mlog}(S^{k-1}). \tag{25}$$

where $\max()$ is element-wise maximization. BiMap, ReEig, and LogEig mimic transformation, non-linear rectified activation, and classification. The architecture of TSMNet+SPDDSMBN (Kobler et al., 2022) can be explained as $f_{tc} \to f_{sc} \to f_{BiMap} \to f_{ReEig} \to f_{SPDDSMBN} \to f_{LogEig}$, where $f_{tc}$ and $f_{sc}$ denote temporal and spatial convolution, and $f_{SPDDSMBN}$ denotes SPD domain-specific momentum batch normalization, which is a SPD batch normalization layer for domain adaptation. For simplicity, we abbreviate TSMNet+SPDDSMBN as TSMNet.

**Datasets and preprocessing: Radar** dataset (Brooks et al., 2019a) contains 3,000 synthetic radar signals. Following the protocol in Brooks et al. (2019a), each signal is split into windows of length 20, resulting in 3,000 covariance matrices of the size $20 \times 20$ equally distributed in 3 classes. **HDM05** dataset (Müller et al., 2007) consists of 2,273 skeleton-based motion capture sequences executed by different actors. Each frame can be represented as a $93 \times 93$ covariance matrix. In line with Brooks et al. (2019a), we remove some under-represented clips and trim the dataset down to 2086 instances scattered throughout 117 classes. **Hinss2021** dataset (Hinss et al., 2021) is a recently released competition dataset containing EEG signals for mental workload estimation. The dataset is employed for two tasks, namely inter-session and inter-subject classification, which are treated as domain adaptation problems. Recently, geometry-aware methods (Yair et al., 2019; Kobler et al., 2022) have demonstrated promising performance in EEG classification. We follow the Python implementation[1] of Kobler et al. (2022) for data preprocessing. In detail, the python package MOABB (Jayaram & Barachant, 2018) and MNE (Gramfort, 2013) are used to preprocess the datasets. The applied steps include resampling the EEG signals to 250/256 Hz, applying temporal filters to extract oscillatory EEG activity in the 4 to 36 Hz range, extracting short segments ($\leq$ 3s) associated with a class label, and finally obtaining $40 \times 40$ SPD covariance matrices.

**Implementation Details:** The original classification in SPDNet and TSMNet is conducted by the LogEig MLR (matrix logarithm+FC+softmax). To ensure a fair comparison, we substitute their LogEig classifiers with our intrinsic SPD MLRs. We use the standard cross-entropy loss as the training objective and optimize the parameters with the Riemannian AMSGrad optimizer (Bécigneul & Ganea, 2018). The network architectures are represented as $[d_0, d_1, \ldots, d_L]$, where the dimension of the parameter in the $i$-th BiMap layer is $d_i \times d_{i-1}$. For the Radar and HDM05 datasets, we adopt a learning rate of $1e-2$, a batch size of 30, and a maximum training epoch of 200. For the Hinss2021 dataset, in line with Kobler et al. (2022), we apply a learning rate of $1e-3$ with a weight decay of $1e-4$, a batch size of 50, and a training epoch of 50. For better comparison, we also implement the AIM-based gyro SPD MLR Nguyen & Yang (2023) to SPDNet and TSMNet, which is named SPDNet+Gyro-AIM or TSMNet+Gyro-AIM. All experiments use an Intel Core i9-7960X CPU with 32GB RAM and an NVIDIA GeForce RTX 2080 Ti GPU.

**Evaluation Methods:** In line with the previous work (Huang & Van Gool, 2017; Kobler et al., 2022), we use accuracy as the scoring metric for the Radar and HDM05 datasets, and balanced

---

[1]https://github.com/rkobler/TSMNet

Table 2: Results of SPDNet with and without SPD MLRs on the Radar and HDM05 datasets.

(a) Radar dataset.

| Methods | | [20,16,8] | [20,16,14,12,10,8] |
|---|---|---|---|
| SPDNet | | 92.88±1.05 | 93.47±0.45 |
| SPDNet+Gyro-AIM | | 94.53±0.95 | 94.32±0.94 |
| SPDNet+$(\theta,\alpha,\beta)$-LEM | (1,1,0) | 93.55±1.21 | 94.60±0.70 |
| | (0.5,1,1) | **95.29±0.61** | **95.31±0.75** |
| SPDNet+$(\theta)$-LCM | (1) | 93.49±1.25 | 93.93±0.98 |
| | (1.5) | 93.07±1.08 | 94.64±0.91 |

(b) HDM05 dataset.

| Methods | | [93,30] | [93,70,30] | [93,70,50,30] |
|---|---|---|---|---|
| SPDNet | | 57.42±1.31 | 60.69±0.66 | 60.76±0.80 |
| SPDNet+Gyro-AIM | | 58.07±0.64 | 60.72±0.62 | 61.14±0.94 |
| SPDNet+$(\theta,\alpha,\beta)$-LEM | (1,1,0) | 56.97±0.61 | 60.69±1.02 | 60.28±0.91 |
| | (0.5,1.0,1/30) | **59.30±0.63** | **62.84±0.50** | **63.06±0.76** |
| SPDNet+$(\theta)$-LCM | (1) | 48.55±2.35 | 47.61±1.82 | 49.10±1.94 |

accuracy (*i.e.,*the average recall across classes) for the Hinss2021 dataset. Ten-fold experiments on the Radar and HDM05 datasets are carried out with randomized initialization and split, while on the Hinss2021 dataset, models are fit and evaluated with a randomized leave 5% of the sessions (inter-session) or subjects (inter-subject) out cross-validation scheme.

**Hyper-parameters:** We implement the SPD MLRs induced by both the parameterized metrics (LEM and LCM) and deformed metrics ($(\theta,\alpha,\beta)$-LEM and $(\theta)$-LCM). Therefore, in our SPD MLRs, we have a maximum of three hyper-parameters, *i.e.,*$\theta,\alpha,\beta$, where $(\alpha,\beta)$ are associated with O$(n)$-invariance and $\theta$ controls deformation. For $(\alpha,\beta)$ in $(\theta,\alpha,\beta)$-LEM, recalling Eq. (2), $\alpha$ is a scaling factors, while $\beta$ measures the relative significance of traces. As scaling is less important (Thanwerdas & Pennec, 2019), we set $\alpha = 1$. We select the value of $\beta$ from the candidate set $\{1, 1/n, 1/n^2, 0, -1/n + \epsilon, -1/n^2\}$, where $n$ is the dimension of input SPD matrices in SPD MLRs[2]. These chosen values for $\beta$ allow for amplifying, neutralizing, or suppressing the trace components, depending on the characteristics of the datasets. For the deformation factor $\theta$, we roughly select its values around the deformation boundary. Specifically, for $(\theta,\alpha,\beta)$-LEM, $(\theta,\alpha,\beta) = (0.5, 1, 0)$ is known as the inverse Euclidean metric. Therefore, the candidate values of $\theta$ in $(\theta,\alpha,\beta)$-LEM consist of $\{0.25, 0.5, 0.75, 1, 1.25, 1.5\}$. For $(\theta)$-LCM, $\theta$ is select from the set $\{0.5, 1, 1.5\}$.

## 6.1 EXPERIMENTAL RESULTS

For each family of SPD MLRs, we report two representative baselines: the standard SPD MLR induced from the standard metric ($\theta = 1, \alpha = 1, \beta = 0$), and the one induced from the deformed metric with selected hyper-parameters. If the standard SPD MLR is already saturated, we only report the results of the standard ones. In Tabs. 2 and 3, we denote $(\theta,\alpha,\beta)$-LEM as the baseline model endowed with the SPD MLR induced by $(\theta,\alpha,\beta)$-LEM. So does $(\theta)$-LCM.

**Radar:** In line with Brooks et al. (2019a), we evaluated our classifiers on the Radar dataset under two network architectures: [20, 16, 8] for the 2-layer configuration and [20, 16, 14, 12, 10, 8] for the 5-layer configuration. The 10-fold results (mean±std) are presented in Tab. 2a. Generally speaking, our SPD MLRs achieve superior performance against the vanilla LogEig MLR. Among all SPD MLRs, the ones induced by (0.5,1,1)-LEM achieve the best performance on this dataset. Although the SPD MLRs induced by standard LEM and LCM are slightly worse than the AIM-based gyro SPD MLR, our SPD MLRs with proper parameters achieve comparable or even better performance than the AIM-based gyro SPD MLR. Moreover, for both $(\theta,\alpha,\beta)$-LEM and $(\theta)$-LCM, the associated SPD MLRs with proper deformation factor $\theta$ outperform the standard SPD MLRs induced by the standard metrics, demonstrating the effectiveness of our parameterization.

**HDM05:** Following Huang & Van Gool (2017), three architectures are evaluated on this dataset: [93, 30], [93. 70, 30] and [93, 70, 50, 30]. On this dataset, the SPD MLR under the standard LCM are already saturated. Besides, the SPD MLRs based on $(\theta)$-LCM exhibit considerably slower convergence. The models fail to converge even after 500 training epochs. Consequently, we report the results after 500 epochs of training for LCM-based SPD MLR. The slow convergence of LCM-based SPD MLR could be attributed to the specific characteristics of the HDM05 dataset, which might interact differently with the $(\theta)$-LCM metric compared to $(\theta,\alpha,\beta)$-LEM. Nevertheless, the SPD MLR induced by $(0.5, 1, 1/30)$-LEM achieves the best performance. Interestingly, although our SPD MLR induced by standard LEM is slightly inferior to AIM-based gyro SPD MLR, the SPD MLR based on $(0.5, 1, 1/30)$-LEM consistently achieves better performance. This phenomenon demonstrates the advantage of our framework's versatility.

---

[2]The purpose of including a small positive constant $\epsilon \in \mathbb{R}_+$ is to ensure O$(n)$-invariance, *i.e.,*$(\alpha,\beta) \in \mathbf{ST}$.

Table 3: Results comparison of TSMNet with and without SPD MLRs on the Hinss2021 dataset.

(a) Inter-session

| Methods | | Balanced Acc. |
|---|---|---|
| SPDDSMBN | | 53.83±9.77 |
| SPDDSMBN+Gyro-AIM | | 53.36±9.92 |
| SPDDSMBN+$(\theta, \alpha, \beta)$-LEM | (1,1,0) | 53.51±10.02 |
| | (0.5,1,0.05) | 55.26±8.93 |
| SPDDSMBN+$(\theta)$-LCM | (1) | 55.71±8.57 |
| | (1.5) | **56.43±8.79** |

(b) Inter-subject

| Methods | | Balanced Acc. |
|---|---|---|
| SPDDSMBN | | 49.68±7.88 |
| SPDDSMBN+Gyro-AIM | | 50.65±8.13 |
| SPDDSMBN+$(\theta, \alpha, \beta)$-LEM | (1,1,0) | 51.41±7.98 |
| | (1.25,1,0) | 52.52±6.83 |
| SPDDSMBN+$(\theta)$-LCM | (1) | 52.93±7.76 |
| | (0.5) | **54.14±8.36** |

**Hinss2021:** Following Kobler et al. (2022), we adopt the architecture of [40,20]. The results (mean±std) of leaving 5% out cross-validation are presented in Tab. 4. Once again, our intrinsic classifiers demonstrate improved performance compared to the baseline in the inter-session and inter-subject scenarios. More interestingly, different from the performance on the HDM05 dataset, SPD MLRs based on $(\theta)$-LCM achieve the best performance (increase 2.6% for inter-session and 4.46% for inter-subject), indicating that this metric can faithfully capture the geometry of data in the Hinss2021 dataset. This finding highlights the adaptability and versatility of our framework, as it can effectively leverage different Riemannian metrics based on the intrinsic geometry of the data.

Table 4: Training efficiency (s/epoch).

| Methods | Radar | HDM05 | Hinss2021 inter-session | inter-subject |
|---|---|---|---|---|
| Baseline | 1.36 | 1.95 | 0.18 | 8.31 |
| MLR-Gyro-AIM | 1.75 | 31.64 | 0.38 | 13.3 |
| MLR-LEM | 1.5 | 4.7 | 0.24 | 8.35 |
| MLR-LCM | 1.35 | 3.29 | 0.18 | 10.13 |

**Model Efficiency:** We adopt the deepest architectures, namely [20, 16, 14, 12, 10, 8] for the Radar dataset, [93, 70, 50, 30] for the HDM05 dataset, and [40, 20] for the Hinss2021 dataset. For simplicity, we focus on the SPD MLRs induced by standard metrics, *i.e.,*LEM and LCM. We also implement AIM-based gyro SPD MLR. The average training time (in seconds) per epoch is reported in Tab. 4. In general, when compared to the AIM-based gyro SPD MLR, LEM- and LCM-based SPD MLRs exhibit superior efficiency, especially when dealing with a larger number of classes. Notably, the HDM05 dataset comprises 117 classes, where the LEM- and LCM-based SPD MLRs require only one-ninth of the training time compared to the AIM-based gyro SPD MLR. This discrepancy can be attributed to the computational complexity of AIM, which involves more matrix decompositions, incurring higher computational costs. In contrast, due to the rapid computation of PEMs, the PEM-based SPD MLR is more computationally efficient. This contrast becomes more obvious when dealing with a huge number of classes, as each class necessitates an SPD parameter, entailing additional Riemannian computations.

## 7 CONCLUSION

In this paper, we provide a general framework for building SPD MLR under any PEM. We extend the existing LEM and LCM into parameterized families of metrics and showcase our framework under these metrics. Our framework also provides an intrinsic explanation for the widely used LogEig classifier. The consistent superior performance in extensive experiments also supports our claims. As a future avenue, our framework can also be readily applied to other kinds of PEMs.

**Limitations:** To develop the SPD MLR based on margin distance to the hyperplane, we only construct SPD MLRs under PEMs, including standard and deformed LEM and LCM. In future work, we aim to extend and develop efficient SPD MLRs to other metrics, such as AIM. We note that though Nguyen & Yang (2023) has implemented AIM-based SPD MLR, their margin distance is pseudo-gyrodistance, which does not really solve Eq. (14).

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

## A NOTATIONS

For better understanding, we briefly summarize all the notations used in this paper in Tab. 5.

Table 5: Summary of notations.

| Notation | Explanation |
| --- | --- |
| $\{\mathcal{M}, g\}$ or abbreviated as $\mathcal{M}$ | A Riemannian manifold |
| $T_P\mathcal{M}$ | The tangent space at $P \in \mathcal{M}$ |
| $g_p(\cdot, \cdot)$ or $\langle \cdot, \cdot \rangle_P$ | The Riemannian metric at $P \in \mathcal{M}$ |
| $\| \cdot \|_P$ | The norm induced by $\langle \cdot, \cdot \rangle_P$ on $T_P\mathcal{M}$ |
| $\mathrm{Log}_P$ | The Riemannian logarithmic map at $P$ |
| $\mathrm{Exp}_P$ | The Riemannian exponential map at $P$ |
| $\Gamma_{P_1 \to P_2}$ | The Riemannian parallel transportation along the geodesic connecting $P_1$ and $P_2$ |
| $H_{a,p}$ | The Euclidean hyperplane |
| $\tilde{H}_{\tilde{A},P}$ | The SPD hyperplane |
| $\odot$ | A Lie group operation |
| $\{\mathcal{M}, \odot\}$ | A Lie group |
| $P_\odot^{-1}$ | The group inverse of $P$ under $\odot$ |
| $L_P$ | The Lie group left translation by $P \in \mathcal{M}$ |
| $f_{*,P}$ | The differential map of the smooth map $f$ at $P \in \mathcal{M}$ |
| $f^*g$ | The pullback metric by $f$ from $g$ |
| $\mathcal{S}_{++}^n$ | The SPD manifold |
| $\mathcal{S}^n$ | The Euclidean space of symmetric matrices |
| $\langle \cdot, \cdot \rangle$ | The standard Frobenius inner product |
| $\| \cdot \|_{\mathrm{F}}$ | The standard Frobenius norm |
| $\mathbf{ST}$ | $\mathbf{ST} = \{(\alpha, \beta) \in \mathbb{R}^2 \mid \min(\alpha, \alpha + n\beta) > 0\}$ |
| $\langle \cdot, \cdot \rangle^{(\alpha, \beta)}$ | The $\mathrm{O}(n)$-invariant Euclidean inner product |
| mlog | Matrix logarithm |
| Chol | Cholesky decomposition |
| $\mathrm{Dlog}(\cdot)$ | The diagonal element-wise logarithm |
| $\lfloor \cdot \rfloor$ | The strictly lower triangular part of a square matrix |
| $\mathbb{D}(\cdot)$ | A diagonal matrix with diagonal elements from a square matrix |
| $\Pi_P$ | The tangential projection at $P$ mapping a Euclidean gradient into a Riemannian one |
| $\nabla_P f$ | The Euclidean gradient of $f$ w.r.t. $P$ |

## B BRIEF REVIEW OF RIEMANNIAN MANIFOLDS

Intuitively, manifolds are locally Euclidean spaces. Differentials are the generalization of classical derivatives. For more details on smooth manifolds, please refer to Tu (2011); Lee (2013). Riemannian manifolds are the manifolds endowed with Riemannian metrics, which can be intuitively viewed as point-wise inner products. When manifolds are endowed with Riemannian metrics, various Euclidean operators can find their counterparts in manifolds. A plethora of discussions can be found in Do Carmo & Flaherty Francis (1992).

**Definition B.1** (Riemannian Manifolds). A Riemannian metric on $\mathcal{M}$ is a smooth symmetric covariant 2-tensor field on $\mathcal{M}$, which is positive definite at every point. A Riemannian manifold is a pair $\{\mathcal{M}, g\}$, where $\mathcal{M}$ is a smooth manifold and $g$ is a Riemannian metric.

The main paper relies on pullback isometry to study SPD manifolds. This idea is a natural generalization of bijection from set theory.

**Definition B.2** (Pullback Metrics). Suppose $\mathcal{M}, \mathcal{N}$ are smooth manifolds, $g$ is a Riemannian metric on $\mathcal{N}$, and $f : \mathcal{M} \to \mathcal{N}$ is smooth. Then the pullback of a tensor field $g$ by $f$ is defined point-wisely,

$$(f^*g)_p(V_1, V_2) = g_{f(p)}(f_{*,p}(V_1), f_{*,p}(V_2)), \tag{26}$$

where $p$ is an arbitrary point in $\mathcal{M}$, $f_{*,p}(\cdot)$ is the differential map of $f$ at $p$, and $V_1, V_2$ are tangent vectors in $T_p\mathcal{M}$. If $f^*g$ is positive definite, it is a Riemannian metric on $\mathcal{M}$, called the pullback metric defined by $f$.

**Definition B.3** (Isometries). If $\{M, g\}$ and $\{\widetilde{M}, \widetilde{g}\}$ are both Riemannian manifolds, a smooth map $f : M \to \widetilde{M}$ is called a (Riemannian) isometry if it is a diffeomorphism that satisfies $f^*\widetilde{g} = g$.

If two manifolds are isometric, they can be viewed as equivalent. Riemannian operators in these two manifolds are also closely related.

A Lie group is a manifold with a smooth group structure. It is a combination of algebra and geometry.

**Definition B.4** (Lie Groups)**.** A manifold is a Lie group, if it forms a group with a group operation $\odot$ such that $m(x, y) \mapsto x \odot y$ and $i(x) \mapsto x_\odot^{-1}$ are both smooth, where $x_\odot^{-1}$ is the group inverse of $x$.

The exponential & logarithmic maps and parallel transportation are also crucial for Riemannian approaches in machine learning. To bypass the notation burdens caused by their definitions, we review the geometric reinterpretation of these operators, introduced in Pennec et al. (2006); Do Carmo & Flaherty Francis (1992). In detail, in a manifold $\mathcal{M}$, geodesics correspond to straight lines in Euclidean space. A tangent vector $\overrightarrow{xy} \in T_x\mathcal{M}$ can be locally identified to a point $y$ on the manifold by geodesic starting at $x$ with initial velocity of $\overrightarrow{xy}$, i.e. $y = \mathrm{Exp}_x(\overrightarrow{xy})$. On the other hand, the logarithmic map is the inverse of the exponential map, generating the initial velocity of the geodesic connecting $x$ and $y$, *i.e.*, $\overrightarrow{xy} = \mathrm{Log}_x(y)$. These two operators generalize the idea of addition and subtraction in Euclidean space. For the parallel transportation $\Gamma_{x \to y}(V)$, it is a generalization of parallelly moving a vector along a curve in Euclidean space. we summarize the reinterpretation in Tab. 6.

Table 6: Reinterpretation of Riemannian Operators.

| Operations | Euclidean spaces | Riemannian manifolds |
|---|---|---|
| Straight line | Straight line | Geodesic |
| Subtraction | $\overrightarrow{xy} = y - x$ | $\overrightarrow{xy} = \log_x(y)$ |
| Addition | $y = x + \overrightarrow{xy}$ | $y = \exp_x(\overrightarrow{xy})$ |
| Parallelly moving | $V \to V$ | $\Gamma_{x \to y}(V)$ |

At last, we briefly review the Riemannian gradient. It is a natural generalization of the Euclidean gradient.

**Definition B.5** (Riemannian gradient)**.** The Riemannian gradient $\tilde{\nabla} f$ of a smooth function $f \in C^\infty(\mathcal{M})$ is a smooth vector field over $\mathcal{M}$, satisfying

$$\langle \tilde{\nabla}_p f, V \rangle_p = V(f), \forall p \in \mathcal{M}, V \in T_p\mathcal{M} \tag{27}$$

## C  SPD HYPERPLANES AS SUBMANIFOLDS OF SPD MANIFOLDS

**Proposition C.1.** *SPD hyperplanes (as defined in Eq.* (12)*) under any geometrically complete Riemannian metric $g$ are submanifolds of SPD manifolds.*

This claim can be proven by either definition (Tu, 2011, Def. 9.1) or the constant rank level set theorem (Tu, 2011, Thm. 11.2). We focus on the latter.

*Proof.* Consider any $P \in \mathcal{S}_{++}^n$ and $A \in T_P\mathcal{S}_{++}^n$. Define the function $f(S) = \langle \mathrm{Log}_P S, A \rangle_P :$ $\mathcal{S}_{++}^n \to \mathbb{R}$. For the SPD hyperplane $\tilde{H}_{A,P}$, we have $\tilde{H}_{A,P} = f^{-1}(0)$. Due to geodesically completeness, $\mathrm{Log}_P$ is globally defined, and $f$ is therefore well-defined. We can rewrite $f$ as a composition, i.e., $f = h \circ \mathrm{Log}_P$, where $h(\cdot) = \langle \cdot, A \rangle_P$ is a linear map.

Since $\mathrm{Log}_P$ is a diffeomorphism, and $h(\cdot)$ is a linear map, the rank of $f$ is globally constant. So there exists a neighbourhood (e.g., the whole SPD manifold) of $f^{-1}(0)$, where the rank of $f$ is constant. According to the constant rank level set theorem (Tu, 2011, Thm. 11.2), we can obtain the claim. $\square$

# D PROOFS FOR THE LEMMAS, PROPOSITIONS, THEOREMS, AND COROLLARIES STATED IN THE PAPER

## D.1 PROOF OF LEM. 3.3

*Proof.* By Thm. 2.2, we have the following,

$$\langle \text{Log}_P Q, A \rangle_P = \langle \phi_{*,P} \phi_{*,\phi(P)}^{-1} (\phi(Q) - \phi(P)), \phi_{*,P} A \rangle \tag{28}$$

$$= \langle \phi(Q) - \phi(P)), \phi_{*,P} A \rangle \tag{29}$$

Therefore, the SPD hyperplane $\tilde{H}_{A_k, P_k}$ corresponds to the Euclidean hyperplane $H_{\phi_{*,P_k}(A_k), \phi(P_k)}$, due to the isometry of $\phi$. Furthermore, the distances to margin hyperplanes are equivalent to the following,

$$\inf_{\phi(Q)} \|\phi(S)) - \phi(Q)\|_F \tag{30}$$

$$\text{s.t.} \langle \phi(Q) - \phi(P_k), \phi_{*,P_k} A_k \rangle = 0. \tag{31}$$

The problem above is the familiar Euclidean distance from a point to a hyperplane. By simple computation, one can obtain the results. □

## D.2 PROOF OF LEM. 3.4

*Proof.* For simplicity, we abbreviate $\odot_\phi$ and $g^\phi$ as $\odot$ and $g$. By abuse of notation, we further denote $Q \odot P_\odot^{-1}$ as $QP^{-1}$, where $P_\odot^{-1}$ is the inversion of $P$ under $\odot$. According to Thm. 2.2, $\{\mathcal{S}_{++}^n, \odot\}$ is an Abelian group, $g$ is bi-invariant Riemannian metric. By Lin (2019, Lem. 6), any parallel transportation can be expressed by a differential of left translation,

$$\Gamma_{P \to Q} = L_{QP^{-1}*, P}, \forall P, Q \in \mathcal{S}_{++}^n. \tag{32}$$

□

## D.3 PROOF OF LEM. 3.5

*Proof.* Due to the geodesic completeness of $\mathcal{S}_{++}^n$, the existence interval of any geodesic is $\mathbb{R}$. Parallel transportation along geodesic thus exists for all $t \in \mathbb{R}$. Through Picard's uniqueness in ODE theories, one can obtain the results. □

## D.4 PROOF OF THM. 3.6

*Proof.*

$$A_k = \Gamma_{I \to P_k}(\tilde{A}_k) \tag{33}$$

$$= \phi_{*,\phi(P_k)}^{-1} \circ \phi_{*,I}(A_k) \tag{34}$$

One can obtain the results by putting Eq. (34) into Eq. (16).

□

## D.5 PROOF OF PROP. 4.1

*Proof.* As shown by Chen et al. (2023a), both LEM and LCM are PEMs. As the pullback of LCM, $(\theta)$-LCM is therefore PEM. Besides, Thanwerdas & Pennec (2023) also indicates that $(\alpha, \beta)$-LEM is isometric to the standard LEM. As the pullback of $(\alpha, \beta)$-LEM, $(\theta, \alpha, \beta)$-LEM is hence PEM. □

## D.6 PROOF OF PROP. 4.2

*Proof.* Let us first review a well-known fact of deformed metrics from Thanwerdas & Pennec (2022). Let $\theta$-$g$ be the deformed metric on SPD manifolds pulled back from $g$ by the power function $(\cdot)^\theta$ and scaled by $\frac{1}{\theta^2}$. Then when $\theta$ tends to 0, for all $P \in \mathcal{S}_{++}^n$ and all $V \in T_P \mathcal{S}_{++}^n$, we have

$$(\theta\text{-}g)_P(V, V) \to g_I(\text{mlog}_{*,P}(V), \text{mlog}_{*,P}(V)). \tag{35}$$

By Eq. (35), we can readily obtain the results. □

### D.7 PROOF OF COR. 4.3

*Proof.* We only need to distinguish the pullback maps of $(\theta, \alpha, \beta)$-LEM and $(\theta)$-LCM. Denoting $\phi(S) = \frac{1}{|\theta|}S^\theta$, then we have:

$$\phi(I) = \frac{1}{|\theta|}I, \tag{36}$$

$$\phi_{*,I}(A) = \text{sgn}(\theta)(A), \forall A \in T_I \mathcal{S}_{++}^n. \tag{37}$$

Next, we begin to derive the pullback maps one by one.

$(\theta, \alpha, \beta)$**-LEM:** We define the following map

$$\psi^{\text{LEM}} = \phi \circ \text{mlog} \circ f \tag{38}$$

where $f : \mathcal{S}^n \to \mathcal{S}^n$ is the linear isometry between the standard Frobenius inner product and the $O(n)$-invariant inner product $\langle\cdot,\cdot\rangle^{(\alpha,\beta)}$. Then $\psi^{\text{LEM}}$ pulls back the standard Euclidean metric on $\mathcal{S}^n$ to $(\theta, \alpha, \beta)$-LEM on $\mathcal{S}_{++}^n$. Putting Eqs. (37) and (38) into Eq. (18), we can obtain the results.

$(\theta)$**-LCM:** Denoting $\psi^{\text{LCM}} = \phi \circ \text{Chol} \circ \text{Dlog}$, then $\psi^{\text{LCM}}$ pulls back the standard Euclidean metric on the Euclidean space $\mathcal{L}^n$ of lower triangular matrices to the $(\theta)$-LCM on $\mathcal{S}_{++}^n$. Putting $\psi^{\text{LCM}}$ and Eq. (38) into Eq. (18), we can obtain the results[3]. $\square$

### D.8 PROOF OF PROP. 5.1

To prove Prop. 5.1, we first present two lemmas about the general cases under PEMs.

One can observe that Eq. (18) and Eq. (19) are very similar to a Euclidean MLR. However, since $\phi$ is normally non-linear and $P_k$ is an SPD parameter, Eq. (18) cannot hastily be identified with a Euclidean MLR. However, under some special circumstances, SPD MLR can be reduced to the familiar Euclidean MLR. To show this result, we first present the Riemannian Stochastic Gradient Descent (RSGD) under PEMs. General RSGD (Bonnabel, 2013) is formulated as

$$W_{t+1} = \text{Exp}_{W_t}(-\gamma_t \Pi_{W_t}(\nabla_W f|_{W_t})) \tag{39}$$

where $\Pi_{W_t}$ denotes the projection mapping Euclidean gradient $\nabla_W f|_{W_t}$ to Riemannian gradient, and $\gamma_t$ denotes learning rate. We have already obtained the formula for the Riemannian exponential map as shown in Eq. (7). We proceed to formulate $\Pi$.

**Lemma D.1.** *For a smooth function $f : \mathcal{S}_{++}^n \to \mathbb{R}$ on $\mathcal{S}_{++}^n$ endowed with any kind of PEMs, the projection map $\Pi_P : \mathcal{S}^n \to T_P \mathcal{S}_{++}^n$ at $P \in \mathcal{S}_{++}^n$ is*

$$\Pi_P(\nabla_P f) = \phi_{*,P}^{-1}(\phi_{*,P}^{-*})(\nabla_P f), \tag{40}$$

*where $\phi_{*,P}^{-*}$ is the adjoint operator of $\phi_{*,P}^{-1}$, i.e.,$\langle V_1, \phi_{*,P}^{-1}V_2\rangle_P = \langle\phi_{*,P}^{-*}V_1, V_2\rangle_P$, for all $V_i \in T_P \mathcal{S}_{++}^n$.*

*Proof.* Given any smooth function $f : \mathcal{S}_{++}^n \to \mathbb{R}$, denote its Riemannian gradient at $P$ as $\tilde{\nabla}_P f \in T_P \mathcal{S}_{++}^n$. Then we have the following,

$$\langle\tilde{\nabla}_P f, V\rangle_P = V(f), \forall V \in T_P \mathcal{S}_{++}^n. \tag{41}$$

By Eq. (4) and canonical chart, we have

$$\langle\phi_{*,P}\tilde{\nabla}_P f, \phi_{*,P}V\rangle = \langle\nabla_P f, V\rangle, \forall V \in T_P \mathcal{S}_{++}^n \cong \mathcal{S}^n, \tag{42}$$

where $\nabla_P f$ is the Euclidean gradient. By the arbitrary of $V$, we have

$$\phi_{*,P}^* \phi_{*,P} \tilde{\nabla}_P f = \nabla_P f, \tag{43}$$

where $\phi_{*,P}^*$ is the adjoint operator of the linear homomorphism $\phi_{*,P}$ w.r.t. $\langle,\rangle$. $\square$

We can describe the special case we mentioned with the above lemma.

---

[3]The differential of Cholesky decomposition is presented in Lin (2019, Prop. 4)

**Lemma D.2.** *Supposing the differential map $\phi_{*,I}$ is the identity map, and $P_k$ in Eq. (18) is optimized by PEM-based RSGD, then Eq. (18) can be reduced to a Euclidean MLR in the codomain of $\phi$ updated by Euclidean SGD.*

*Proof.* Define a Euclidean MLR in the codomain of $\phi$ as

$$p(y = k \mid S) \propto \exp(\langle \phi(S) - \bar{P}_k, \bar{A}_k \rangle)), \text{ with } \bar{P}_k, \bar{A}_k \in \mathcal{S}^n. \tag{44}$$

We call this classifier $\phi$-EMLR.

Define the SPD MLR under the PEM induced by $\phi$ is

$$p(y = k \mid S) \propto \exp(\langle \phi(S) - \phi(P_k), \tilde{A}_k \rangle), \text{ with } P_k \in \mathcal{S}^n_{++}, \tilde{A}_k \in \mathcal{S}^n. \tag{45}$$

Supposing the SPD MLR and $\phi$-EMLR satisfying $\bar{P}_k = \phi(P_k)$. Other settings of the network are all the same, indicating the Euclidean gradients satisfying

$$\frac{\partial L}{\partial \bar{P}_k} = \frac{\partial L}{\partial \phi(P_k)}. \tag{46}$$

The updates of $\bar{P}_k$ in the $\phi$-EMLR is

$$\bar{P}'_k = \bar{P}_k - \gamma \frac{\partial L}{\partial \bar{P}_k}. \tag{47}$$

The updates of $P_k$ in the SPD MLR is

$$P'_k = \mathrm{Exp}_{P_k}(-\gamma \Pi_{P_k}(\nabla_{P_k} f)) \tag{48}$$

$$= \phi^{-1}(\phi(P_k) - \gamma \phi^{-*}_{*,P_k} \frac{\partial L}{\partial P_k}) \tag{49}$$

Therefore $\phi(P'_k)$ satisfies

$$\phi(P'_k) = \phi(P_k) - \gamma \phi^{-*}_{*,P_k} \frac{\partial L}{\partial P_k} \tag{50}$$

$$= \phi(P_k) - \gamma \phi^{-*}_{*,P_k} \phi^{*}_{*,P_k} \frac{\partial L}{\partial \phi(P_k)} \tag{51}$$

$$= \phi(P_k) - \gamma \frac{\partial L}{\partial \phi(P_k)} \tag{52}$$

$$= \bar{P}'_k \tag{53}$$

Eq. (51) comes from the Euclidean chain rule of differential. Let $Y = \phi(X)$, then we have

$$\frac{\partial L}{\partial Y} : \mathrm{d}\, Y = \frac{\partial L}{\partial Y} : \phi_{*,X}\, \mathrm{d}\, X = \phi^{*}_{*,X} \frac{\partial L}{\partial Y} : \mathrm{d}\, X, \tag{54}$$

where : means Frobenius inner product.

The equivalence of $\bar{A}_k$ and $\tilde{A}_k$ is obvious. By natural induction, the claim can be proven. $\square$

Now, We can directly prove Prop. 5.1 by Lem. D.2.

