# OpenReview forum: "Riemannian Multiclass Logistics Regression for SPD Neural Networks"
_ICLR.cc/2024/Conference — ICLR 2024 Conference Withdrawn Submission_

### Official Review · Reviewer_myc6 · 2023-10-30

**Soundness:** 3 good
**Presentation:** 3 good
**Contribution:** 2 fair
**Rating:** 5
**Confidence:** 3

**Summary:**

It introduces and elaborates on Pullback Euclidean Metrics (PEMs) and their significance in understanding SPD matrices. The research also explores the application of SPD manifold learning in recognition tasks, referencing datasets like Radar and HDM05. Furthermore, the paper presents the deformed Log-Euclidean Metric (LEM) and Log-Cholesky Metric (LCM) and provides rigorous proofs to support its theoretical constructs, suggesting a blend of deep mathematical exploration with practical applications.

**Strengths:**

Originality: The paper introduces and delves deep into the mathematical intricacies of SPD manifolds, a topic that is not commonly addressed in mainstream machine learning literature. The introduction of Pullback Euclidean Metrics (PEMs) and their properties showcases a fresh perspective on the subject.

Quality: The technical depth of the paper is commendable. The authors have ensured rigorous mathematical formulations and have backed their claims with appropriate proofs and derivations. The experiments, especially those involving SPDNet on datasets like Radar and HDM05, further validate the paper's claims.

Clarity: While the content is dense, the paper is structured in a logical manner, allowing readers familiar with the domain to follow the progression of ideas. The use of mathematical notations and diagrams aids in understanding the complex concepts presented.

Significance: The exploration of SPD manifolds in the context of machine learning holds potential significance for specialized applications. The paper's findings could pave the way for more advanced techniques in the domain, especially for those who are looking to bridge the gap between pure mathematics and practical machine learning applications.

Incorporating these strengths, the paper demonstrates a solid contribution to the field, especially for those interested in the deeper mathematical underpinnings of machine learning techniques.

**Weaknesses:**

Complexity of Proofs: The paper provides detailed proofs for lemmas, propositions, theorems, and corollaries, such as Lem. 3.3, Lem. 3.4, Lem. 3.5, and Thm. 3.6. While this depth is commendable, it might make the content less accessible to readers unfamiliar with the specific mathematical constructs. Simplifying or providing more intuitive explanations alongside these proofs could make the paper more accessible.

Generalization of Metrics: The paper introduces deformed metrics like (θ, α, β)-LEM and (θ)-LCM. While these are interesting constructs, the paper might benefit from a clearer justification or motivation for introducing these deformations. What specific problems or limitations with the standard LEM and LCM are these deformations addressing?

Empirical Validation: While the paper showcases results on datasets like Radar and HDM05, it might benefit from a broader set of experiments on diverse datasets to validate the universality of the proposed methods. Additionally, comparisons with state-of-the-art methods in the same domain would provide a clearer picture of the paper's contributions.

Hyperparameter Discussion: The paper mentions hyperparameters like θ, α, β in the context of the deformed metrics. A more in-depth discussion or analysis on the sensitivity of these hyperparameters and their impact on the results might be beneficial. This would give readers insights into the robustness of the proposed methods and how sensitive they are to parameter tuning.

Potential Over-reliance on Mathematical Constructs: The paper heavily leans on mathematical constructs and proofs. While this is essential for a rigorous presentation, the paper might benefit from more real-world examples or intuitive explanations to bridge the gap between theory and practical applications.

Clarity on Novelty: While the paper introduces concepts like Pullback Euclidean Metrics (PEMs) and their significance, it might benefit from a clearer delineation of what exactly is novel in this work compared to existing literature. Explicitly highlighting the gaps in current research and how this paper fills those gaps can strengthen its contributions.

Potential for Broader Applications: The paper focuses on SPD manifold learning in recognition tasks. Exploring or discussing potential applications in other domains or tasks could enhance the paper's appeal to a broader audience.

**Questions:**

Clarification on Deformed Metrics: What was the primary motivation behind introducing the deformed metrics like (θ, α, β)-LEM and (θ)-LCM? Are there specific limitations or challenges with the standard LEM and LCM that these deformations address?

Empirical Validation Scope: The results on datasets like Radar and HDM05 are presented. Could the authors provide insights into why these specific datasets were chosen? Are there plans to test the proposed methods on a broader set of datasets?

Comparison with State-of-the-Art: How does the proposed method compare with the current state-of-the-art methods in SPD manifold learning? Are there benchmark results available that can provide a clearer picture of the paper's standing in the current research landscape?

Hyperparameter Sensitivity: How were the hyperparameters like θ, α, β chosen for the experiments? Is there any empirical analysis available on their sensitivity and impact on the results? This would be crucial for understanding the robustness of the proposed methods.

Intuitive Explanations: Given the depth of mathematical constructs and proofs, could the authors provide more intuitive explanations or real-world examples to make the content more accessible to a broader audience?

Novelty Delineation: It would be beneficial if the authors could explicitly highlight the novel contributions of this work in comparison to existing literature. What are the specific gaps in current research that this paper addresses?

Broader Applications: Are there potential applications of the proposed methods beyond SPD manifold learning in recognition tasks? If so, could the authors discuss or explore these in the paper to enhance its appeal?

Potential for Extensions: Given the introduction of Pullback Euclidean Metrics (PEMs), are there plans or ideas for extending this concept further in future research?

Reproducibility: For the benefit of the research community, could the authors provide details on the availability of code and datasets? This would aid in reproducibility and further exploration by other researchers.

Limitations and Future Work: Every research has its limitations. Could the authors shed light on the potential limitations of their proposed methods and any plans for addressing them in future work?

**Details Of Ethics Concerns:**

The content does not include any ethics concerns.

---

### Official Review · Reviewer_vpbo · 2023-10-31

**Soundness:** 2 fair
**Presentation:** 2 fair
**Contribution:** 1 poor
**Rating:** 3
**Confidence:** 5

**Summary:**

This paper proposes utilizing LEM and LCM while training multiclass logistic regression models on SPD manifolds. The proposed methods were examined in radar recognition, human action recognition, and electroencephalography (EEG) classification tasks.

**Strengths:**

The paper studies different metrics for classification on SPD manifolds.  The theoretical discussions provide nice insights covering different metrics extending the current solutions proposed in the literature.

**Weaknesses:**

In the experimental analyses, the proposed methods perform on par with baseline and slightly outperforms the baseline in some cases depending on the hyper-parameters. To show the novelty of the proposed method and its superiority compared to the baseline and related work more clearly, experimental analyses should be extended with additional datasets and backbones.

Also, although theoretical discussions on different formulation of the metrics are nice, they should be extended considering their complexity (both theoretical and experimental analyses of memory footprints and running time complexity) and equivalence properties.

**Questions:**

-	In the experimental analyses, the proposed methods perform on par with the baseline and in some cases, it outperforms the baseline. Therefore, they are pretty sensitive to hyperparameter selection. How should the hyperparameters be selected? More precisely, hyperparameter selection procedure was discussed on page 8. In addition to the setup, could you please also explain which algorithms should be used to search for these parameters, and how do selected hyperparameters generalize?

-	For different datasets, different hyperparameters provide the best accuracy. What can the reason be?

-	How does the accuracy change for different optimizers? Can you provide learning curves for different optimizers?

-	Can you theoretically/experimentally compare footprints of the proposed methods and the baseline/sota?

---

### Official Review · Reviewer_fcxe · 2023-11-01

**Soundness:** 3 good
**Presentation:** 3 good
**Contribution:** 2 fair
**Rating:** 3
**Confidence:** 4

**Summary:**

This paper proposes generalizations of multiclass logistic regression (MLR) to SPD manifolds. In order to simplify the derivation of such generalizations, the authors consider SPD manifolds associated with famillies of Log-Euclidean and Log-Cholesky metrics. Experiments are performed on radar recognition, human action recognition, and electroencephalography classification for evaluating the proposed approach.

**Strengths:**

* Theoretical results are justified
* The proposed method enhances the performance of some existing SPD neural networks

**Weaknesses:**

* The main paper's weakness is lack of novelty in developing SPD MLR
* The proposed method lacks of motivation
* Experimental results are weak w.r.t. to state-of-the-art methods
* Experiments are not enough to validate the benefit of the proposed method

**Questions:**

I believe that the main contribution of the paper is the construction of MLR on SPD manifolds associated with variants of Log-Euclidean and Log-Cholesky metrics. However, as I said above, the main paper's weakness is lack of novelty in developing SPD MLR. Its key ingredient here is the reformulation of MLR from the perspective of distance to hyperplanes. This formulation has been used by Ganea et al. (2018) in the context of hyperbolic neural networks, and by Nguyen & Yang (2023) in the context of SPD neural networks. The difference between the proposed reformulation and the one in Nguyen & Yang lies in the definition of the distance from an SPD matrix to an SPD hyperplane in Equation (14). However, this definition is a direct consequence of Theorems 2.23 and 2.24 in Nguyen & Yang (2023), where it is stated that the pseudo-gyrodistance is equal to the SPD gyrodistance (which turns out to be the geodesic distance) in the case of Log-Euclidean and Log-Cholesky metrics. Thus there is no finding in Equation (14).

It is noted in Xavier Pennec et al. (2006) that the Log-Euclidean framework turns SPD manifolds into flat spaces. Thank to this property,
extensions of classical methods in Euclidean space to SPD manifolds pose no challenges. Sereval examples are given in Xavier Pennec et al. (2006). Please see also Arsigny et al. (2005) (reference below). This is also the case of Log-Cholesky framework, where it has been shown (Lin, 2019) that the sectional curvature of SPD manifolds under this framework is constantly zero (thus SPD manifolds become flat spaces).

In summary, Equation (14) is a direct consequence of Theorems 2.23 and 2.24 in Nguyen & Yang (2023), and the Log-Euclidean and Log-Cholesky frameworks that the paper focusses on turn SPD manifolds into flat spaces and therefore allow one to straightforwardly derive extensions of MLR to SPD manifolds (Lemma 3.3 and Corollary 4.3).

In terms of experimental results, although the proposed method achieve improvements when it is used in the frameworks of some SPD neural networks, it is hard to see the real benefit of the proposed method in comparison with state-of-the-art methods. For instance, the method in Li et al. (2018) (reference below) achieves 84.47 $\pm$ 1.52 on HDM05 compared to 63.06 $\pm$ 0.76 obtained by the proposed method. The same issue applies to experimental evaluations for radar and EEG classification where more efforts are needed to demonstrate the efficacy of the proposed method compared to state-of-the-art methods. Also, there is no comparison against SPDNetBN which has been shown to outperform SPDNet in a variety of problems.

**Questions:**
1. Can the proposed method be used in SPDNetBN ? If yes, how about the performance of the resulting network ?

**References**

1. Arsigny, V., Fillard, P., Pennec, X., and Ayache, N. Fast and Simple Computations on Tensors with Log-Euclidean Metrics. Technical Report RR-5584, INRIA, 2005
2. Li, C., Cui, Z., Zheng, W., Xu, C., Ji, R., and Yang, J.: Action-Attending Graphic Neural Network. IEEE Trans. Image Process. 27(7): 3657-3670 (2018)

---

### Official Review · Reviewer_9moa · 2023-11-09

**Soundness:** 3 good
**Presentation:** 3 good
**Contribution:** 3 good
**Rating:** 5
**Confidence:** 4

**Summary:**

This work is concerned about developing deep neural networks for manifold-valued data. First, it extends multiclass logistics regression for arbitrary pullback Euclidean metrics (PEM) on SPD manifolds. Second, it develops multiclass logistics regression for the deformed log-Euclidean metric and log-Cholesky metric, which are two specific examples of PEM. Additionally, this work uses the proposed framework to explain LogEig multiclass logistics regression previously proposed in the literature. Experimental study is conducted on three datasets to show its advantage over the existing relevant methods.

**Strengths:**

1. Developing deep neural networks for manifold-valued data is of importance and worth exploring.
2. This work puts forward a general framework for multiclass logistics regression for arbitrary pullback Euclidean metrics (PEM) on SPD manifolds. Under this framework, multiclass logistics regression is developed for two specific PEMs. Experimental study shows the improved recognition performance.
3. This work is well supported by theoretical results in the field of Riemannian manifold, which is welcome.

**Weaknesses:**

1. The motivation and the benefit of this work can be made more clear. For example, why does the gyro-structures lead to "limiting the generality" and what sense of generality is meant here? Also, why does the proposed one have more flexibility? Clarifying these issues will help.

2. Listing the differences from gyro-structure in Section 3 is welcome. Meanwhile, the advantages of the proposed approach over the gyro-structure based one shall be clearly stated. Also, in Sec. 3.1 and 3.2, please discuss the challenges or difficulties of developing the proposed generalisation, which will help to highlight the technical contribution of this work.

3. Please clarify the relationship between Sections 3 and 4. For example, which one is more essential for the improved performance obtained in the experimental study? Does the utilisation of the deformed LEM and LCM have to rely on the proposed framework?

4. Experimental study needs to be strengthened.
    1) The covariance matrix extracted from the there datasets are essentially man-made manifold data. For the HDM05 dataset related to motion capture, applying deep learning based methods to the raw data may be able to achieve better performance than manually extracting SPD matrices as features. Therefore, it will be better to demonstrate this work with truly manifold-valued data that cannot be effectively handled by the existing deep learning neural networks.
    2) Experimental result can be better explained. For example, the poor performance of ($\theta$)-LCM in Table 2 (b). The explanation provided in the last paragraph on page 8 is vague.
    3) Experimental study is limited in terms of the number of methods in comparison and the scale of the datasets.
    4) How are the proper values of $\theta$, $\alpha$ and $\beta$ selected in the experimental study? For example, via multi-fold cross-validation of training set or directly on the test set? Please make this clear.

5. Considering the complexity of the theoretical concepts introduced in this work, more graphical illustrations can be used to help with explaination.

**Questions:**

In the context of the comments in "Weaknesses" column, please answer the following questions:

1. Please describe the motivation of this work.
2. Please describe its advantages over the gyro-structure based framework.
3. What are the challenges or difficulties of developing the proposed generalisation?
4. What leads to the poor performance of ($\theta$)-LCM in Table 2 (b)?
3. How are the proper values of $\theta$, $\alpha$ and $\beta$ selected in the experimental study?

**Details Of Ethics Concerns:**

No concerns.